# ASPECT-BASED SENTIMENT CLASSIFICATION VIA REINFORCEMENT LEARNING

## ABSTRACT

Aspect-based sentiment classification aims to predict sentimental polarities of one or multiple aspects in texts. As texts always contain a large proportion of task-irrelevant words, accurate alignment between aspects and their sentimental descriptions is the most crucial and challenging step. State-of-the-art approaches are mainly based on word-level attention learned from recurrent neural network variants (e.g., LSTM) or graph neural networks. From another view, these methods essentially weight and aggregate all possible alignments. However, this mechanism heavily relies on large-scale supervision training: without enough labels, it could easily overfit with difficulty in generalization. To address this challenge, we propose SentRL, a reinforcement learning-based framework for aspect-based sentiment classification. In this framework, input texts are transformed into their dependency graphs. Then, an agent is deployed to walk on the graphs, explores paths from target aspect nodes to their potential sentimental regions, and differentiates the effectiveness of different paths. By limiting the agent's exploration budget, our method encourages the agent to skip task-irrelevant information and focus on the most effective paths for alignment purpose. Our method considerably reduces the impact of task-irrelevant words and improves generalization performance. Compared with competitive baseline methods, our approach achieves the highest performance on public benchmark datasets with up to $3.7\%$ improvement.

## 1 INTRODUCTION

The goal of aspect-based (also known as aspect-level) sentiment classification is to predict the sentiment polarities of individual aspects. As shown in Figure 1, given a sentence *"I like this computer but do not like the screen"*, the sentiment of the aspect *"computer"* is positive because of *"like"*. Meanwhile, the sentiment of the aspect *"screen"* is negative for *"do not like"*.

Aspect-based sentiment classification is challenging, where the core problem is to correctly align aspects with their sentiment descriptions. State-of-the-art methods rely on supervision signals to automatically learn such alignment. By leveraging textual context and word-level attention learned from deep models (Vo & Zhang, 2015; Dong et al., 2014; Bahdanau et al., 2014; Luong et al., 2015; Xu et al., 2015; Wang et al., 2016; Tang et al., 2016b; Ma et al., 2017a; He et al., 2018; Zhang et al., 2018; 2019; Gao et al., 2019; Tang et al., 2020), existing methods have made great progress on discovering aspect-specific sentimental statements.

Meanwhile, the existing methods could suffer serious overfitting problems, as natural language inevitably includes a large proportion of task-irrelevant texts, or noise from the perspective of machine learning. Ideally, with a sufficient amount of training labels, the existing methods could effectively contain the negative impact of such task-irrelevant information. In practice, because of the high variance in language expression, it is costly to collect a large number of task-specific labels, and it is difficult to guarantee the expected label sufficiency. With limited labels, the existing approaches could easily include task-irrelevant information into decision processes, overfit training data, and end up with inferior generalization performance to unseen data.

To effectively reduce the impact of task-irrelevant information, we propose SentRL, a reinforcement learning based framework for aspect-level sentiment classification. In our approach, input texts are firstly transformed into graph objects (e.g., dependency graphs (Covington, 2001)), where nodes are words and edges indicate syntactic dependencies/relations between them. Next, we deploy a

Figure 1: Dependency graph of a given sentence. **Blue** words are the aspects. **Green** and **red** are *"positive"* and *"negative"* sentiment respectively. The dependency graph effectively reduces the distances between aspects and sentimental descriptions and avoids polysemy words (e.g., *"like"*). In our approach, an agent is deployed to walk from the aspect word to the sentimental regions, which avoids task-irrelevant information and achieves more effective and efficient performance.

policy-based agent to discover aspect-related sentiment descriptions in the graphs. This agent is geared with a language understanding module so that it is able to update exploration states and make sentiment decisions for individual aspects. Unlike existing methods that aggregate potential sentiment information from all possible textual contexts or words, our agent strives to leverage the most relevant exploration paths under a limited budget. This strategy not only requires the agent to focus on the most effective paths but also encourages the agent to skip task-irrelevant regions. Using standard back-propagation methods, the policy network and the language understanding module are jointly trained. From public benchmark datasets, we observe our method could achieve up to $3.6\%$ improvement compared with competitive state-of-the-art methods. The main contributions of our work are listed below.

- A novel reinforcement learning framework for aspect-based sentiment classification is proposed. It accurately pinpoints the most effective path between sentiment descriptions and the target aspects, and effectively avoids the impact of the task-irrelevant regions.

- A policy network is developed to provide an agent with exploration guidance. This network iteratively provides suggestions on next-hop selection. In particular, the framework is permutation invariant and guarantees the consistency and reliability of the model.

- A language understanding module is developed to help an agent "remember" its exploration history and make the final sentiment prediction.

- Extensive experiments on representative benchmark datasets are evaluated. The results demonstrate the effectiveness, efficiency, and robustness of our approach.

## 2 RELATED WORK

### 2.1 ASPECT-BASED SENTIMENT CLASSIFICATIONS

Aspect-based sentiment classification is to identify sentiment polarities of one or more aspects in given texts (Thet et al., 2010). Aspects could be either substantial objects (e.g., *computer* and *car*) or conceptional objects (e.g., *service* and *atmosphere*). There are usually three sentiment categories including *positive*, *neutral*, and *negative*, while more sophisticated categories could be explored. Conventional approaches (Kiritchenko et al., 2014) treat input texts as word sequences, and deploy separate feature extraction modules as well as classification modules. Deep learning-based sentiment analysis methods (Tang et al., 2016a) take contextual information regarding the word order into consideration by using LSTM (Hochreiter & Schmidhuber, 1997; Liu et al., 2018). Attention-based approaches are proposed (Tang et al., 2016c; Bahdanau et al., 2014; Luong et al., 2015; Xu et al., 2015; Wang et al., 2016; Tang et al., 2016b; Ma et al., 2017a; Huang & Carley, 2019; Ma et al., 2017b; Huang et al., 2018; Li et al., 2018) to improve the effectiveness of contextual feature extraction. While such approaches utilize sequential models and attention mechanisms to learn features from word sequences, they could require a large amount of training labels to be well generalized for natural language expressions with non-trivial variance (*e.g.,* long sentences with majority of irrelevant contextual words).

## 2.2 SENTIMENT ANALYSIS ON GRAPHS

Syntactic dependency tree (Covington, 2001) is a widely adopted data structure that encodes syntactic dependencies between words in input texts. To this end, aspect-based sentiment classification problems can be cast into node classification problems on syntactic dependency graphs. Existing methods develop variants of graph convolutional neural networks (Yao et al., 2019; Linmei et al., 2019; Zhang et al., 2019; Ghosal et al., 2019; Bai et al., 2020; Wang et al., 2019) to extract contextual features that represent aspect nodes. However, such methods still suffer the following issues. First, due to the complexity of natural language, these models require a large amount of training data to be well generalized. Second, such methods (Kipf & Welling, 2016; Yao et al., 2019; Linmei et al., 2019) usually perform under transductive settings with limited application scopes.

## 2.3 REINFORCEMENT LEARNING IN NLP

In the domain of NLP, reinforcement learning (RL) is usually explored in interactive tasks (Wang et al., 2018), such as text-based games (Narasimhan et al., 2015b). Recently, RL methods are also developed for complex tasks, such as relation extraction (Narasimhan et al., 2015a; Qin et al., 2018), image captioning (Pasunuru & Bansal, 2017), popularity prediction (He et al., 2016; Zhou & Wang, 2018), coreference resolution (Yin et al., 2018), and reasoning in question answering (Xiong et al., 2017; Wu et al., 2018). Unlike previous works, we are the first to study RL methods that intelligently collect information from syntactic dependency graphs for aspect-based sentiment analysis.

## 3 OUR APPROACH

In this work, we propose a reinforcement learning-based framework, SentRL, for solving the challenges. The details are introduced below:

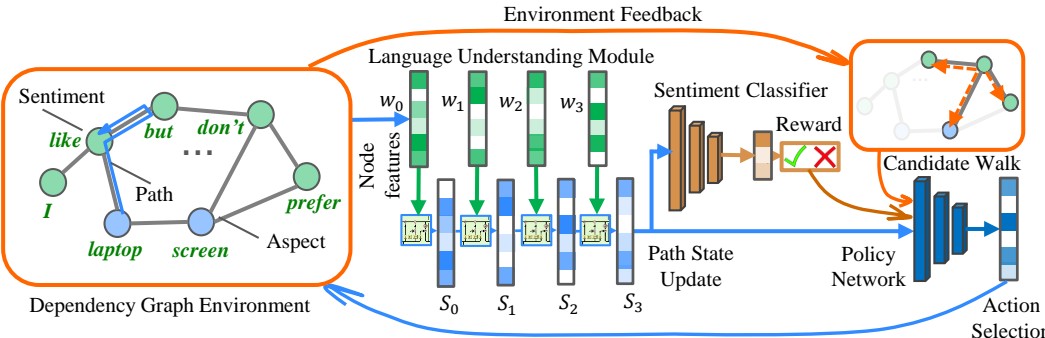

Figure 2: Framework of our SentRL method. A dependency graph is first obtained. The related sentiment clues could be more close in dependency graph compared with the original sentence space. An agent starts from an aspect node and walk though the graph edges. The walk direction is determined by a policy network which utilizes the current walk state and the direction candidates for making the decision. A semantic understanding module is used to update path states and a sentiment classifier is designed to have the final sentiment prediction based on the walk state.

## 3.1 PRELIMINARY

Given the target texts $c = \{w_1^c, w_2^c, w_3^c, ..., w_n^c\}$, where $w_i^c$ represents the $i$-th word in $c$. $n$ is the total number of the words. The target aspect of $c$ is denoted as $a_c = \{w_{r+1}^c, ..., w_{r+m}^c\}$, where $r+1$ indicates the start location of the aspect and $m$ is the length of the aspect. The aspect is either a single-word format (e.g., *"computer"*, *"service"*, and *"screen"*) or multiple-word format (e.g., *"HDMI port"* and *"sport mode"*). There could be one or more aspects in $c$, and different aspects could have different sentimental categories. The goal of aspect-based sentiment classification is to predict sentiment polarity (i.e., *"positive"*, *"neutral"*, and *"negative"*) for each given aspect.

## 3.2 Dependency Graph Extraction

A dependency graph $G_c = (V(G_c), E(G_c))$ is extracted from $c$. $V(G_c)$ indicates all vertices/nodes in $G_c$, and each vertice corresponds to a word in $c$. $E(G_c)$ denotes the edges of $G_c$. Each edge in $G_c$ represents the syntactical relation categories between a pair of words. By deploying the dependency graph, we transfer the sequential text data into graph-structured format. In $G_c$, the long-distance pair of words in texts could be close in the graph. This provides extra syntactical knowledge and makes down-stream algorithms easier to pinpoint the sentimental words.

## 3.3 Path Searching via Reinforcement Learning

Compared with state-of-the-art approaches utilizing graph learning algorithms (e.g., GCN, attention, and RNN), we proposed a reinforcement learning framework to explore the most effective aspect-sentiment path on the dependency graph. First, an agent starts walking from the target aspect node. Then, a policy network selects the most effective walk based on the previous walk history and all feasible walk candidates. Next, a semantic understanding module is deployed to aggregate the path state and a sentiment classifier is used to obtain the final sentiment prediction. In summary, there are three components in our framework, 1) A **Semantic Understanding Module** which aggregates the comprehensive walk state, 2) A **Policy Network** which makes the walk decision, and 3) A **Sentiment Classifier** which obtains the final prediction of the sentiment categories. The details of each component are introduced in the following sections.

## 3.4 Semantic Understanding Module

The semantic understanding module aims to provide a comprehensive state of the walk for 1) allowing the policy network to make the effective walk action and 2) letting the sentiment classifier obtain a final prediction. There is a requirement that the state updating mechanism should make the walking procedure as a Markov Decision Process (MDP). The expression is shown below:

$$P(S_{t+1}|S_0, A_0, ..., S_t, A_t) = P(S_{t+1}|S_t, A_t), \tag{1}$$

where $S_i$, $A_i$, and $R_i$ are the state, action, and reward of the $i$-th move respectively. Eq. equation 1 indicates that the path state of $(t + 1)$-th move, $S_{t+1}$, should be only relevant to the current state $S_t$ and action $A_t$, and irrelevant to earlier states. While, the information of the entire path is crucial and necessary for sentimental analysis tasks. To this end, $S_t$ is required to preserve both the current and all previous walk information.

To achieve this goal, we deploy the general Long Short-Term Memory (LSTM) Hochreiter & Schmidhuber (1997) structure in our framework, as LSTM and its varieties have been well validated as an effective way to capture both the feature and the sequential knowledge in a given sample. In our model, the function is shown below:

$$S_t = \text{LSTM}(A_t, S_{t-1}), \tag{2}$$

where $S_t$ is the current state, which could also be considered as the hidden state updated in each loop. $A_t$ is the node feature of the corresponding $t$-th action (i.e., word embedding) which is walked though by agent in the $t$-th move. Semantic understanding module keeps updating $S_t$ for each walk. We consider $S_t$ is effective enough to aggregate and preserve the entire path information.

## 3.5 Policy Network

The goal of the policy network is to guide the agent to find the most effective paths between aspect nodes and sentiment description in the obtained dependency graph. Specifically, it is designed to select the action based on the previous walk path and the next move candidates. Since the semantic understanding module effectively captures the knowledge of existing path, another crucial clue is the candidates nodes of the next walk.

However, there are two unique challenges for deploying conventional policy network in graph-structured data. First, the policy network should perceive all possible candidates for making the most effective action. While, general deep network structure requires consistent data format as input (e.g., images and videos with consistent resolution). In $G_c$, each node could have a different

number of neighbor (1-hop) nodes associated with various connection categories. Thus, it is challenging to define an action space which covers all the candidate paths in the same output. Second, the policy network should be permutation irrelevant to the candidate input, which means the action of the agent would not be affected by the order of the candidates.

To this end, we proposed a structural input mechanism to address these challenges. When the agent has arrived at a node, it obtains all the connected nodes (including the previously walked nodes and itself). We assume the set of the nodes are $V = \{v_1, v_2, ..., v_m\}$, where $m$ is the number of all candidate nodes around $i$-th nodes. We concatenate current state $S_t$ with each node $s_i, i = \{1, 2, ..., m\}$ and obtain a vector. We put the vector into the policy network to get a candidate score of the node $v_i$. The expression is formulated as shown below:

$$s_i = \text{policy}(\text{cat}(S_t, v_i)), \quad i = \{1, 2, ..., m\}, \tag{3}$$

where $\text{cat}(\cdot)$ indicates the concatenation operation. This strategy considers the probability of each candidates separately and it relaxes the input inconsistency challenge while still preserves the local structural information of the dependency graph. When the scores of all candidates are obtained, the agent would go to the node which has the highest score. The function is shown below:

$$a = \arg\max_{i=\{1,2,...,m\}} (s_i). \tag{4}$$

In general sentiment analysis tasks, the aspect is usually a single word while there are a few aspects containing multiple words (e.g., "*HDMI port*"). In this scenario, we merge all the edges of the corresponding nodes together and consider the aspect as a single node. A consistent walk length is set to stop the walk process. In our experiments, we set walk length of 3 which is effective enough for achieving high performance. If the distances are not equal to 3, we observed that if the distances are shorter, the repetitive path exists around the final aspect (e.g., walk through *"good"* to *"good"*). Even if the consistent walk-length is not enough, the agent still tries to explore the most likely path for accurate prediction. More details are discussed in Appendix C

## 3.6 SENTIMENT CLASSIFIER

After the agent finishes a path, a sentiment classifier is deployed to obtain the final sentiment prediction. After the last move, we consider $S_t$ is able to provide the comprehensive information of the entire path. To this end, we directly forward $S_t$ into a sentiment classifier. The expression of the sentiment classifier is shown below:

$$p = \delta(W_p S_t + b_p), \tag{5}$$

where $\delta(\cdot)$ is the non-linear activation. In our implementation, we deploy Softmax activation to predict one polarity from the candidate pool. $W_p$ is the weights and $b_p$ is the bias.

## 3.7 REWARD

The reward provides the information which directly guides the training procedure. Conventional RL frameworks have a clear reward design, which lets the policy network performs well in each stage. However, in our framework, the ultimate goal is high classification performance which relays on multiple modules (i.e., Semantic Understanding Module, Policy Network, and Sentiment Classifier). To this end, we directly consider the correct prediction as the reward for all the modules. We deploy Mean Squared Error (MSE) as the accuracy reward and the function is shown below:

$$r_{acc} = -\|y - p\|_2^{\text{F}}, \tag{6}$$

where $r_{acc}$ is the accuracy reward, $y$ is the ground truth label. $r_{acc}$ would only be calculated once after the walk is finished. It is a straightforward yet effective reward. In the training procedure, all the three modules are simultaneously optimized to achieve the best performance. Other evaluation metrics (e.g., Cross Entropy) could also be deployed. We empirically evaluate different metrics, and we found most metrics are effective while MSE achieves the best performance.

## 3.8 IMPLEMENTATION & OPTIMIZATION

In our implementation, we use a 2-layer fully-connected (FC) neural network to parameterize the policy network that maps the state vector $S_t$ to a probability distribution over all possible actions.

ReLU activation is deployed in the first layer while no activation in the second layer. However, to obtain a smooth score across different candidates, the sigmoid function is deployed across all actions. The goal of this step is to make the optimization procedure more smooth and robust.

In the optimization procedure, some strategies are used to improve its efficiency and effectiveness. For the policy network, simply choosing the candidate with the highest score is a non-differentiable sampling strategy making it infeasible to calculate the weight gradients. Gumbel Softmax Jang et al. (2017) employs a continuous distribution to approximate a non-differentiable sample. In this way, we can have a one-hot vector as the output of the agent while still preserving the parameter gradients. Meanwhile, Gumbel Softmax also fully explores the distribution of candidate scores and add more diversity to the training process. The core functions of Gumbel Softmax is shown as follows:

$$z_i = \frac{exp((log(\pi_i) + g_i)/\tau)}{\Sigma_{j=1}^{k} exp((log(\pi_j) + g_j)/\tau)} \tag{7}$$

where $g_i$ are $i.i.d$ samples drawn from $Gumbel(0, 1)$ distribution, $\pi_i$ are the class probabilities, $z_i$ are the generated samples, $\tau$ is the temperature parameter. The smaller the $\tau$, the final result is more close to a one-hot vector. In the training process of our implementation, we directly apply Gumbel Softmax in PyTorch. The output would be a one-hot vector, while the gradients are calculated from the Gumbel Softmax. For the testing process, since we do not need to calculate the gradients and add diversity, we simply use $argmax(\cdot)$ function to get the one-hot vector as the output.

## 4 EXPERIMENTS

### 4.1 DATASETS

We consider five public benchmark datasets on aspect-based sentiment analysis (Yoo & Gretzel, 2008). Table 1 illustrates their basic information, including label distribution and training-testing sample split settings. A brief description of the datasets is as follows.

Table 1: Statistics of the evaluation datasets

| Dataset | Lap14 | | Twitter | | Rest14 | | Rest15 | | Rest16 | |
| Split | Train | Test | Train | Test | Train | Test | Train | Test | Train | Test |
| --- | --- | --- | --- | --- | --- | --- | --- | --- | --- | --- |
| Positive | 994 | 341 | 1561 | 173 | 2164 | 728 | 912 | 326 | 1240 | 469 |
| Neutral | 464 | 169 | 3127 | 346 | 637 | 196 | 36 | 34 | 69 | 30 |
| Negative | 870 | 128 | 1560 | 173 | 807 | 196 | 256 | 182 | 439 | 117 |

- **LAP14** (Pontiki et al., 2014) includes review/descriptions of various laptops from users, where human annotators tagged both aspects and their polarities.
- **TWITTER** (Dong et al., 2014) contains mentions in tweets on a diverse types of aspects, including celebrities (*e.g., Bill Gates*), products (*e.g., xBox*), and companies (*e.g., Google*), where their sentiment labels are manually annotated.
- **REST14** (Pontiki et al., 2014) focuses on restaurants' reviews, where aspect terms and their polarities are annotated.
- **REST15** (Pontiki et al., 2015) includes reviews for laptops, restaurants, and hotels, where annotators identify opinions expressed towards specific entities and their attributes.
- **REST16** (Pontiki et al., 2016) also includes reviews on restaurants, where each review may contain multiple aspects.

We follow the commonly adopted experimental setting described in (Tang et al., 2016c). One sample will be removed if it is associated with conflicting polarities or it contains unclear aspects. For fair comparisons, identical word representations are adopted in SentRL and baseline methods: A 300-dimensional pretrained GloVe model (Pennington et al., 2014) is further fine-tuned by a bidirectional LSTM (Zhang et al., 2019). The dimensionality of the hidden state in SentRL is set to 300 in all the cases. ADAM optimizer (Kingma & Ba, 2014) is used to train SentRL with a learning rate of 0.0005.

## 4.2 BASELINES

We consider the following state-of-the-art approaches as baselines in this study. While a more detailed description is included in Appendix A, a brief overview is as follows.

- **SVM**. Kiritchenko et al. (2014) deploys in-house sequence tagger to detect aspect terms with SVMs serving as the classifiers, where input texts are modeled as naturally-ordered word sequences.

- **LSTM**. Tang et al. (2016a) extends conventional LSTM (Hochreiter & Schmidhuber, 1997) to predict sentiment polarity, where input texts are modeled as naturally-ordered word sequences.

- **MemNet**. Tang et al. (2016c) uses external memories and a multi-hop architecture to learn the importance of context words for aspect-sentiment representations, where input texts are modeled as naturally-ordered word sequences.

- **AOA**. Huang et al. (2018) leverages the idea of attention-over-attention for sentiment classification, where input texts are modeled as naturally-ordered word sequences.

- **IAN**. Ma et al. (2017b) employs attention networks to learn contextual representations for aspects, where input texts are modeled as naturally-ordered word sequences.

- **TNet-LF**. Li et al. (2018) develops a target-specific transformation to obtain context information of aspects, where input texts are modeled as naturally-ordered word sequences.

- **ASGCN**. Zhang et al. (2019) considers the contextual information by utilizing dependency graphs, and a GCN and an attention mechanism are used to learn sentimental representations. Note that input texts are modeled as graphs of words with syntactic structures.

We employ the same evaluation metrics discussed in (Zhang et al., 2019). In particular, evaluation results are obtained by averaging 3 runs with the random initialization, where accuracy (ACC) and macro-F1 (F1) are adopted as the metrics. In our implementation, dependency graphs are obtained from spcCy [1], while SentRL is compatible with various syntactic parsers.

Table 2: Sentiment classification performance over public benchmark datasets

| Methods | Lap14 | | Rest14 | | Rest15 | | Rest16 | | Twitter | |
|---|---|---|---|---|---|---|---|---|---|---|
| | ACC | F1 | ACC | F1 | ACC | F1 | ACC | F1 | ACC | F1 |
| SVM (Kiritchenko et al., 2014) | 70.49 | N/A | 80.16 | N/A | N/A | N/A | N/A | N/A | 63.40 | 63.30 |
| LSTM (Tang et al., 2016a) | 69.28 | 63.09 | 78.13 | 67.47 | 77.31 | 55.17 | 86.80 | 63.88 | 69.56 | 67.70 |
| MemNet (Tang et al., 2016c) | 70.64 | 65.17 | 79.61 | 69.64 | 77.31 | 58.28 | 85.44 | 65.99 | 71.48 | 69.90 |
| AOA (Huang et al., 2018) | 72.62 | 67.52 | 79.97 | 70.42 | 78.17 | 57.02 | 87.50 | 66.21 | 72.30 | 70.20 |
| IAN (Ma et al., 2017b) | 72.05 | 67.38 | 79.26 | 70.09 | 78.54 | 52.65 | 84.74 | 55.21 | 72.50 | 70.81 |
| TNet-LF (Li et al., 2018) | 74.61 | 70.14 | 80.42 | 71.03 | 78.47 | 59.47 | **89.07** | 70.43 | 72.98 | 71.43 |
| ASCNN (Zhang et al., 2019) | 72.62 | 66.72 | 81.73 | 73.10 | 78.47 | 58.90 | 87.39 | 64.56 | 71.05 | 69.45 |
| ASGCN-DT (Zhang et al., 2019) | 74.14 | 69.24 | 80.86 | 72.19 | 79.34 | 60.78 | 88.69 | 66.64 | 71.53 | 69.68 |
| ASGCN-DG (Zhang et al., 2019) | 75.55 | 71.05 | 80.77 | 72.02 | 79.89 | 61.89 | 88.99 | 67.48 | 72.15 | 70.40 |
| SentRL | **75.55** | **71.48** | **82.05** | **73.40** | **80.07** | **65.54** | 87.82 | **71.17** | **73.12** | **71.64** |

## 4.3 PERFORMANCE ANALYSIS

The overview of classification performance is reported in Table 2. We observe that SentRL outperforms all the baseline methods in 4 out of 5 cases in terms of accuracy (ACC). In terms of macro-F1, SentRL outperforms in all the cases, with up to 3.7% improvement over the best baseline method.

In addition, to further confirm the effectiveness of SentRL, we visualize the learned sentimental representations and compare it with the ones generated from the second best model, ASGCN (Zhang et al., 2019) by t-SNE method (Maaten & Hinton, 2008). As shown in Figure 3, red, green, and blue dots denote *"negative"*, *"neutral"*, and *"positive''* aspect-based sentiment, respectively. We observe that our approach could better separate different sentiment. Especially for *"neutral"* and *"negative"* ones, our approach could better differentiate them.

---

[1] https://spacy.io/

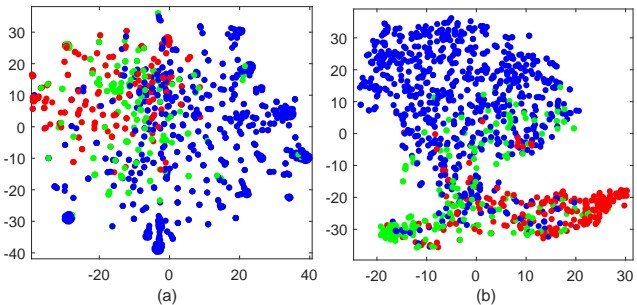

Figure 3: t-SNE visualization of the learned sentimental representations, where (a) is AS-GCN (Zhang et al., 2019) method and (b) is our approach. Different color denotes different sentimental categories.

## 4.4 CASE STUDY

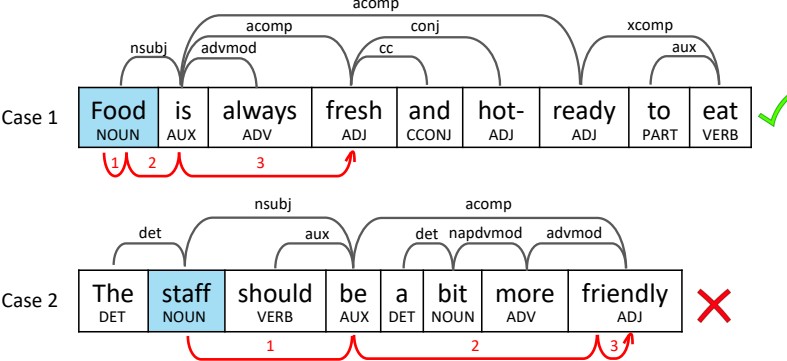

Figure 4: A case study demonstrating traces from aspect words to their related sentimental words: Case 1 is a sample correctly predicted by SentRL, while case 2 is an incorrect prediction. Grey edges denote extracted syntactic dependencies, and red ones denote traces generated by a learned agent. For example, in case 1, the agent trace is *Food→Food→is→fresh*.

We present a case study and visualize traces from aspect words to their related sentimental words discovered by SentRL. One correct (case 1) and one incorrect prediction (case 2) are shown in Figure 4, where words in blue boxes are aspects, grey edges denote syntactic dependencies between words, and red edges are traces left by a learned agent. In case 1, the agent starts from the aspect word *food* and ends at the word *fresh*, collecting necessary information within three hops as expected. In case 2, we notice the agent has a hard time to make a correct prediction. Ideally, starting from the aspect word *staff*, within three hops, a well-learned agent may collect words such as *friendly* and *more*, which may provide minimal evidence that lead the agent to a correct prediction. However, in our investigation, we find there are no similar statements (*e.g., should be a bit more*) that expresses such negative sentiment in the training data. To this end, without accessing commonsense knowledge, it could still be difficult for SentRL to make the correct prediction in case 2.

## 5 CONCLUSION

In this paper, we propose SentRL, a novel aspect-based sentiment classification approach via reinforcement learning. By investigating input texts from their syntactic structures (*e.g.,* dependency graphs), we are able to effectively reduce input variance introduced by diverse expressions in natural languages. On top of the syntactic structures, an agent is deployed to discover the most effective paths that link aspects with their sentimental descriptions. By collecting evidences from the discovered paths, a semantic understanding module in SentRL learns to make the accurate sentimental classification. All the modules in SentRL are simultaneously trained in an end-to-end fashion. Extensive experiments and case studies demonstrate the effectiveness of our method.

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

## A  BASELINES

We compare our model with several baselines methods and state-of-the-art models for a fair and comprehensive comparison. The detailed baseline methods are listed below:

- **SVM** (Kiritchenko et al., 2014) deploys traditional feature extraction methods and uses an in-house sequence tagger to detect aspect terms. Supervised classifiers are used to do the classification.

- **LSTM** (Tang et al., 2016a) proposes an extension on conventional LSTM (Hochreiter & Schmidhuber, 1997) model to predict sentiment polarity. It incorporates target information to fully explore the relevant context words.

- **MemNet** (Tang et al., 2016c) uses external memories and a multi-hop architecture to learn the importance of the context words and further help learning the text representation.

- **AOA** (Huang et al., 2018) introduces attention-over-attention (AOA) in the machine translation field to the sentiment classification. It takes both text-to-aspect and aspect-to-text into consideration.

- **IAN** (Ma et al., 2017b) applies the attention network to both aspects and context to learn the representations interactively. The concatenation of text and context feature is both used to obtain the sentimental classification results.

- **TNet-LF** (Li et al., 2018) introduces a Target-Specific Transformation (TST) component to get the context information of the aspect text. Then it proposes Context-Preserving Transformation (CPT) layers to further learn the abstract feature which is useful for classification tasks.

- **ASGCN** (Zhang et al., 2019) considers the contextual information by utilizing the dependency graph as prior knowledge. A GCN associated with an attention mechanism framework is used to extract most related sentimental representations.

## B  PARAMETER SENSITIVITY

Walk length of the agent is the crucial parameter in our model. In this experiment, we tested the model performance with different walk length (i.e., from 2 to 5) and the result is shown in Figure 5. We observed that our model achieves the best performance when walk length is 3. We assume that if walk length is less than 3, the agent cannot reach to the sentiment word and the performance decreases significantly. Meanwhile, if walk length is more than 3, the performance only have slight decline, and we conjecture this is due to the unnecessary walks which aggregate extra information. The result illustrates the requirement of long enough walk length of our approach, and it also should parameter insensitivity when walk length achieves the threshold.

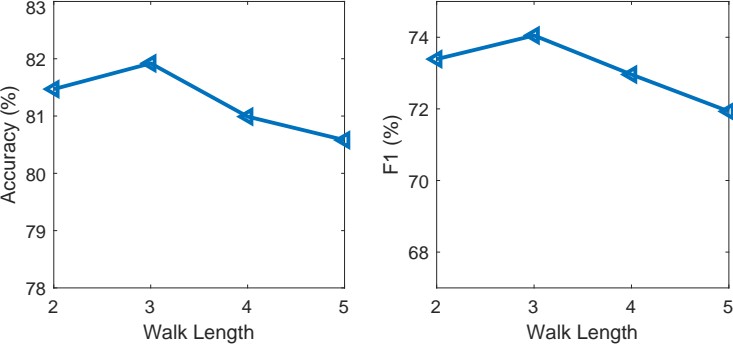

Figure 5: Classification performance with different walk length.

Table 3: Sentiment classification performance based on random walk

| Walk Length | Lap14 | | Rest14 | | Rest15 | | Rest16 | | Twitter | |
|---|---|---|---|---|---|---|---|---|---|---|
| | ACC | F1 | ACC | F1 | ACC | F1 | ACC | F1 | ACC | F1 |
| 1 | 65.88 | 60.05 | 74.37 | 61.51 | 74.11 | 51.59 | 84.30 | 61.13 | 67.34 | 63.86 |
| 2 | 67.47 | 63.28 | 75.44 | 64.82 | 74.76 | 53.15 | 84.44 | 63.29 | 67.34 | 65.17 |
| 3 | 70.42 | 65.10 | 76.03 | 65.89 | 74.91 | 53.85 | 84.98 | 64.12 | 67.93 | 66.12 |
| 4 | 70.51 | 65.43 | 76.15 | 66.28 | 75.61 | 54.04 | 85.15 | 65.00 | 68.14 | 66.82 |
| Ours | **75.55** | **71.48** | **82.05** | **73.40** | **80.07** | **65.54** | **87.82** | **71.17** | **73.12** | **71.64** |

## C  MODEL ANALYSIS

To further analysis the effectiveness of our reinforcement learning module, we intentionally let the agent randomly walk though the dependency graph while keeping the other modules unchanged, and the result is illustrated in Table 3 where the walk length is set from 1 to 4. From Table 3, we observed that when walk length is 1, the performance significantly drops. However, it is still higher than random guess. When the walk length increases, the performance cannot increase consistently. From the results, we could conclude that 1) our reinforcement learning approach is effective and essential for obtaining high performance, and 2) the aspect node does have bias for different sentiment polarity, and walking through the dependency graph could more effectively capture the syntactical knowledge and improve the performance. However, compare with the specifically trained agent based on reinforcement learning, it is not enough for getting high accuracy predictions.

