# OpenReview forum: "Aspect-based Sentiment Classification via Reinforcement Learning"
_ICLR.cc/2021/Conference — Reject_

### Official Review · AnonReviewer2 · 2020-10-25
**Interesting work, but needs more ablation and comparison with recent work**

**Rating:** 5
**Confidence:** 4

**Review:**

Summary:
The paper addresses aspect-based sentiment analysis by running reinforcement learning on the dependency parse of input sentences. The agent learns a policy network to select the most effective walk along edges in the dependency graphs, starting from the target aspect in the input sentences. The state representation is learned with an LSTM. At the end of the path, a sentiment classifier predicts the distribution of the polarity. The reward is the mean squared error between the class label (i.e. sentiment polarity) and the probability predicted by the model. The paper claims that by limiting the agent's budget, the approach forces the agent to discard irrelevant information and focus on the effective paths, enabling the approach to perform well with a small number of training examples.

Strengths:
- The intuition behind the approach of an agent learning dependency paths that can be used for sentiment analysis is appealing. The paper evaluates the performance on 5 data sets, and compares against baselines such as MemNet, TNet-Lf, and ASGCN. Among the baselines, ASGCN also leverage on dependency parses of the input sentences.

Weaknesses:
- The paper failed to compare against a number of publications that also uses dependency parses as input to improve aspect based sentiment analysis, e.g., [1], [2], [3], [4] and [5]. [1] and [2] were referenced in the paper, but their results were not included in the comparison. The results in these 5 papers seem considerably better than in the current submission: e.g., for laptops, a number of these papers obtain over 74% F1 compared to the 71.48% in this submission.

- It seems that there is a big advantage in using dependency parses for sentiment analysis (e.g., [1,2,3,4,5]), so it is unclear how much of the good performance in the current submission is actually from reinforcement learning. While the intuition behind finding dependency paths is appealing,  the ablation that tries to show this is limited to a few examples in the paper. It would be interesting to have a baseline that uses the dependency parse but not reinforcement learning. They could also try and compile some statistics of the relations on the most selected dependency paths, or evaluate a small sample of selected paths with human evaluators.

[1] Syntax-Aware Aspect Level Sentiment Classiﬁcation with Graph Attention Networks
Binxuan Huang and Kathleen M. Carley. EMNLP 2019

[2] Exploiting Typed Syntactic Dependencies for Targeted Sentiment Classiﬁcation Using Graph Attention Neural Network. Xuefeng Bai, Pengbo Liu, and Yue Zhang. ArXiv, submitted on 22nd Feb 2020

[3] Relational graph attention network for aspect-based sentiment analysis. Kai Wang, Weizhou Shen, Yunyi Yang, Xiaojun Quan, and Rui Wang. ACL 2020.

[4] Dependency graph enhanced dual transformer structure for aspect-based sentiment classification. Hao Tang, Donghong Ji, Chenliang Li, and Qiji Zhou. ACL 2020.

[5] Modelling context and syntactical features for aspect based sentiment analysis. Minh Hieu Phan and Philip O. Ogunbona. ACL 2020.

---

> ### Author Response · Authors · 2020-11-25
> **Reply to Reviewer 2**
>
> Thank you very much for reviewing and providing such valuable and frank comments on our work. The suggestions help us a lot in improving our paper. Below are our detailed replies:
>
> $\bullet$ This paper failed to compare against a number of publications
>
> Thanks a lot for your suggestions. We will definitely discuss all the baselines in our revised version. In this paper, we aim to provide a novel paradigm for sentiment classification with less exploration of unnecessary/global knowledge. We consider that this reinforcement learning-based approach is a more human-like mechanism to explore knowledge and eliminate irrelevant information. We hope the reviewer could recognize the novelty of this point.
>
> $\bullet$ It is unclear how much of the good performance in the current submission is actually from reinforcement learning. It would be interesting to have a baseline that uses the dependency parse but not reinforcement learning.
>
> It is a good point. While, consider the reinforcement learning is the crucial module in our approach which cannot be simply removed. To reasonably evaluate the performance, we did two ablation experiments. First, we trained a classifier only based on the aspect node without the walking step. Second, we set the agent to randomly walk in the dependency graph. And the results are shown in the Appendix C.
>
> We observed that when walk length is one, the performance significantly drops while it is still higher than random guess. When the walk length increases, the performance cannot increase consistently. From the results, we could conclude that 1) our reinforcement learning approach is effective and essential for obtaining high performance, and 2) the aspect node does have bias for different sentiment polarity, while it is not enough for getting correct predictions. We added more discussions in our revised version.

---

### Official Review · AnonReviewer3 · 2020-10-28

**Rating:** 5
**Confidence:** 3

**Review:**

The paper proposes an approach to aspect-based sentiment classification, which is the task of identifying the sentiment of a specific phrase or entity in a sentence. The paper proposes to do this by first generating a dependency parse of the entire sentence and then using an RL agent to walk the dependency tree starting from the word or phrase to be classified. The state for the RL agent is the LSTM representation of its history. The final state is then used to classify the sentiment of the aspect.
The approach is evaluated on multiple benchmark datasets for the task and compared with many previously proposed approaches. Most of the compared methods don’t make use of the dependency graphs but the authors also compare with one recent graph convolution-based approach that also makes use of the dependency graph. They show improvements over these methods on most of the benchmarks.

Strong Points
- The approach makes use of dependency graph information in an interesting way to learn to find paths that would lead to better aspect-based sentiment classification.
- The proposed method is well explained and easy to follow.
- The results show improvements on many datasets.

Weak Points
- While the paper compares with many baselines, it doesn’t compare with some highly related work [1,2] which uses Transformer text encoders. Ideally, the paper should discuss these and potentially directly compare with these.
- Due to the use of RL, the proposed approach will be more computationally expensive. It will be good to also discuss this and contrast with other methods.

Questions for authors:
- How do you obtain the dependency tree for the sentences? Have you analyzed how errors in dependency parsing affect performance?
- Can you comment on the computation time for the proposed and competing approaches?
- Can you compare and contrast your approach to [1] and [2]? It would be useful to see if using BERT token embeddings as input embeddings for the proposed method still provides competitive performance as these approaches[1,2] report better performance than the proposed approach.
- Most of the datasets used in the experiments are very small. Given the high sample complexity requirement of RL methods, it is a bit surprising that the RL-based approach works at all for this problem. Can you explain if you found this to be an issue in your experiments?

Suggestions:
- It will be good to also mention the word representations and the text encoder (LSTM or something else) that is used in the compared baselines.
- For path length comparisons in Appendix, please add a pointer to it in the experiments section. It will also be good to see the performance at path length 1 (which wouldn’t require any agent walk).
- Many citations are not properly formatted. Instead of “By leveraging textual context and word-level attention learned from deep models Vo & Zhang (2015); Dong et al. (2014)”, it should be cited as “By leveraging textual context and word-level attention learned from deep models (Vo & Zhang, 2015; Dong et al., 2014)”

[1] Target-Dependent Sentiment Classification With BERT https://ieeexplore.ieee.org/stamp/stamp.jsp?tp=&arnumber=8864964

[2] Dependency Graph Enhanced Dual-transformer Structure for Aspect-based Sentiment Classification https://www.aclweb.org/anthology/2020.acl-main.588.pdf

---

> ### Author Response · Authors · 2020-11-25
> **Reply to Reviewer 3**
>
> Thank you very much for reviewing and providing such valuable and frank comments on our work. The suggestions help us a lot in improving our paper. Below are our detailed replies:
>
> $\bullet$ How do you obtain the dependency tree for the sentences? Have you analyzed how errors in dependency parsing affect performance?
>
> Thanks for these great questions. In our implementation, the dependency graph is obtained from spcCy (https://spacy.io/), while we consider our model is compatible with various syntactic parsers.
> We did not do the dependency graph quality analysis since the ground-truth of the dependency graphs are not available. However, we admit that the dependency graph provides crucial structural knowledge and the accurate dependency parsing is important.
>
> $\bullet$ Can you comment on the computation time for the proposed and competing approaches?
>
> It is a good question. In summary, we consider our approach has similar computation time while considerably smaller space complexity compared with other methods. Specifically, in our model, the agent starts walking from the aspect node and only “see” and “evaluate” the next move based on its neighbor (1-hop) nodes. This strategy does not require exploring the whole sentence. To this end, only a small portion of the dependency graph is involved in the computation process. However, other competing methods (e.g., GCN, attention, and BiLSTM) require to explore the complete input. If the input sentences are big (multiple paragraphs), then both the space and time complexity would become larger than our approach.
>
> $\bullet$ Can you compare and contrast your approach to [1] and [2]?
>
> Thank you very much for this suggestion. We carefully analyzed the papers and both of them are great work. We contacted the authors for their implementation codes but have not got any reply yet. We will add the results when the code is ready.
>
> For paper [1], we further checked their experimental setup and the reported performances (Table 5 and Table 6). While, we found out that except Laptop dataset, [1] deploys the different training/testing splits in the experiments which includes more training samples for Restaurant (3693 vs. 3057) and Twitter (6248 vs. 5495) datasets. In addition, we utilized GloVe word embedding (without fine-tuning the weights) instead of BERT embedding with fine-tuning strategy. To this end, we consider the direct comparison is not applicable without the codes.
>
> For paper [2], we tried to further evaluate the performance of [2]. However, we can not open the provided link in [1] to access the code. We discussed the paper and will compare it when the code is available.
>
>
> $\bullet$ Given the high sample complexity requirement of RL methods, it is a bit surprising that the RL-based approach works at all for this problem. Can you explain if you found this to be an issue in your experiments?
>
> Many thanks for this great question. We did have concerns for well-train a policy network based on the relatively small-scale datasets. While, the experimental results illustrate that our approach is robust enough to handle these datasets.
>
> We admitted that the reinforcement learning strategy is harder to train compared with other general deep learning-based or GCN-based methods. In our experiments, we observed that our approach requires higher CPU running time to process and collect the path information and node representations in the training procedure. In addition, since the agent in our model only explores its neighbor (1-hop) nodes which provide limited global information, our model requires more epochs to converge and achieves the highest performance.
>
> However, our model is more efficient in the testing procedure. The agent only explores the neighbor nodes which does not need the exploration for the complete sentences. To this end, the computational cost, especially the space complexity is lower than other methods.
>
> $\bullet$ Many citations are not properly formatted.
>
> Thank you very much for carefully checking our paper. We double checked and revised all the citation issues in our paper.

---

### Official Review · AnonReviewer1 · 2020-10-28
**Interesting RL implementation but with limited contribution.**

**Rating:** 3
**Confidence:** 5

**Review:**

This paper proposes a reinforcement-learning-based model for aspect-based sentiment analysis. The RL model trains an agent that tries to walk through the most effective path from the aspect target to determine the final sentiment towards this target.

In general, the RL model based on the dependency structure could be deemed as a pioneer implementation on aspect-based sentiment analysis. And this approach is able to remove the negative effect brought by irrelevant context. However, there are a few limitations that need to be addressed:
1. The writing needs to be improved. There are many grammatical mistakes that affect the readability, e.g., "To effectively contain the impact from task-irrelevant information..." from the last paragraph in page 1: the misusing of word "contain". "The goal of aspect-based sentiment classification is to predict sentiment polarities (i.e., “positive”, “neutral”, and “negative”) for each given aspect.": "sentiment polarities"->"sentiment polarity".
2. The proposed model is missing many details. For example, when will the policy terminate? It seems from (3) and (4) there could be infinite loop of candidate actions. And what is exactly the policy network in (3)?
3. The RL model in this case is very limited in terms of exploration. From the construction, the candidate actions only contain connected nodes at each timestamp. This may miss important information, e.g., the second aspect in figure 1 could not traverse to "do not". How do you solve this issue?
4. I am not convinced by the statement that the proposed model is more generalizable compared to baseline models. Where does the generalization ability come from? And I conjecture RL needs even more training data to perform well.
5. As mentioned in the main text, the reward function (6) could be any form, but no experiments are conducted on other forms. The experimental result in Table 2 misses other recent baselines, e.g., "Dependency Graph Enhanced Dual-transformer Structure for Aspect-based Sentiment Classification" that actually outperforms this method.

---

> ### Author Response · Authors · 2020-11-25
> **Reply to Reviewer 1**
>
> We thank the reviewer’s comments and suggestions. We replied all the concerns and please see the details below:
>
> $\bullet$ Paper writing should be improved.
>
> Thanks for pointing it out. Sure, we will further proof read and polishing our paper for the final version.
>
> $\bullet$ Some model details are missing. 1) When will the policy terminate? 2) What is exactly the policy network?
>
> Thanks for the detailed analysis of our model. We agree that these details could make our paper become more comprehensive. Consider the space limitation, we briefly introduced our implementation in Section 3.8, while we added more details and created Section X for all the implementation details.
>
> Specifically, for question 1), it is a reasonable question. In our model, we let the agent walk a consistent length and then terminate the walk process. Since the dependency graph effectively shortens the distance between the aspect node and the sentiment node, we empirically found out that a walk length of 3 is effective enough for achieving high performance. In Appendix Section B, we illustrated the performance with different walk-length (i.e., 2, 3, 4, and 5), and our approach achieves relatively stable performance when walk-length is 3.
>
> We agree that the distances are not equal to 3 for all the cases. In our experiments, we observed that if the distances are shorter, the repetitive path exists around the final aspect (i.e., walk through “good” to “good”). Even if the consistent walk-length is not enough, the agent still tries to explore the most likely path for accurate prediction.
>
> There are some potential solutions such as setting up a stop action to terminate the walk process. While we found the current strategy is the most efficient one with less complicity. We discussed this point in our revised version.
>
> For question 2), the policy network is a 2-layer fully-connected neural network. The non-linear ReLU activation is deployed in the first layer, and the output layer contains only one output which is the score of each candidate next move.
>
> $\bullet$ The candidate actions only contain connected nodes at each timestamp.
>
> It is a good question. We agree that a correct dependency graph provides crucial dependency knowledge for our model to effectively pinpoint the sentiment nodes. While, we consider this concern is out of the scope of our model. In this work, our model aims to search the most related information (i.e., sentiments) based on the structural data (e.g., dependency tree or other prior knowledge).
>
> While it is still a reasonable concern if the dependency graph is not accurate. In this scenario, we could utilize some potential methods such as multi-hop, enhance dependency tree and so on. We discussed it in our current version and will explore this point in our future work.
>
> $\bullet$ Where does the generalization ability come from?
>
> It is a good question. Please let us explain more for this point. In general natural language, the same meaning could be expressed by a lot of different ways, while these expressions/sentences considerably enlarge the variance in feature space which is a challenge for down-stream NLP tasks. While, we assume that the dependency tree could provide clear dependency knowledge and make the expression more similar with less variance in graph space. We experientially empirically tested our approach and it works well.
>
> $\bullet$ No experiments are conducted on other forms. The experimental result in Table 2 misses other recent baselines
>
> Thank you for this suggestion. We tried to further evaluate the performance of [1]. However, we can not open the provided link in [1] to access the code. We discussed the paper and will compare it when the code is available.

---

### Decision · Program_Chairs · 2021-01-07
**Final Decision**

**Decision:**

Reject

**Comment:**

In this paper, the authors proposed a reinforcement-learning-based model for aspect-based sentiment analysis. As raised by the reviewers, 1) the writing needs to be improved: e.g., presenting the details of the proposed method clearly, citing the references properly, etc. 2) related methods need to be implemented for comparison, 3) the reported results are not SOTA compared with existing methods. Moreover, some technical claims are not convincing, which need to be stated more carefully.

In summary, based on its current shape, this paper is not ready to be published in ICLR.